# Antibody-Mediated Rejection and Recurrent Primary Disease: Two Main Obstacles in Abdominal Kidney, Liver, and Pancreas Transplants

**DOI:** 10.3390/jcm10225417

**Published:** 2021-11-19

**Authors:** Tsukasa Nakamura, Takayuki Shirouzu

**Affiliations:** 1Department of Organ Transplantation and General Surgery, Kyoto Prefectural University of Medicine, Kajii-cho 465, Kamigyo-ku, Kyoto 602-8566, Japan; 2Molecular Diagnositcs Division, Wakunaga Pharmaceutical Co., Ltd., 13-4 Arakicho, shinjyuku-ku, Tokyo 160-0007, Japan; shirouzu_t@wakunaga.co.jp

**Keywords:** antibody-mediated rejection, recurrent primary disease, renal transplantation, liver transplantation, pancreas transplantation

## Abstract

The advances in acute phase care have firmly established the practice of organ transplantation in the last several decades. Then, the next issues that loom large in the field of transplantation include antibody-mediated rejection (ABMR) and recurrent primary disease. Acute ABMR is a daunting hurdle in the performance of organ transplantation. The recent progress in desensitization and preoperative monitoring of donor-specific antibodies enables us to increase positive outcomes. However, chronic active ABMR is one of the most significant problems we currently face. On the other hand, recurrent primary disease is problematic for many recipients. Notably, some recipients, unfortunately, lost their vital organs due to this recurrence. Although some progress has been achieved in these two areas, many other factors remain largely obscure. In this review, these two topics will be discussed in light of recent discoveries.

## 1. Introduction

From the late 20th century to the beginning of the 21st century, significant progress has been achieved in acute phase care for transplant patients. These advances firmly place organ transplantation into firmly established therapeutic procedures for organ failure patients. However, the better outcomes in the acute phase become, the more other issues are exposed. Firstly, donor specific anti-human leukocyte antigen (HLA) antibodies (DSA), resulting in chronic antibody-mediated rejection (ABMR), are recognized as major obstacles that we have yet to conquer. Organ transplantation is haunted by DSA unless we change graft sources or develop a new technology. Secondly, controlling the recurrence of the primary disease will continue to be a major issue in many diseases as long as we continue to use live organs and not machines.

The primary purpose of this review is to deepen the understanding of these two issues and to improve graft survival and patient survival after the acute phase of transplants.

## 2. Methods

We have written this review by focusing on two major issues: ABMR due to de novo DSA (dnDSA) and recurrence of primary disease. In preparing this review, English-language abstracts cited in PubMed were selected. Citations were chosen based on their relevance to each section. In Section 3, articles related with dnDSA, not preforming DSA, were selected. As recent dnDSA studies in kidney transplantation are summarized in a table, the most recent randomized trials or prospective cohort studies using representative drugs were selected as much as possible. Since the number of related studies is scarce in liver and pancreas transplants, we selected studies that included therapeutic approaches regardless of study design. In Section 4, we sought studies of a relatively large scale in order to show reliable recurrence data and also basic mechanisms as to why the primary disease recurs, if available.

## 3. An Overview of Antibody-Mediated Rejection

Rejection after organ transplantation is roughly divided into cell-mediated and antibody-related (humoral) immune mechanisms. Originally, T cell-mediated rejections (TCMRs) and ABMR against the ABO blood group were recognized as major barriers. In the 1970s, the introduction of cyclosporin, followed by tacrolimus, mycophenolate mofetil, anti-CD25 antibodies (Abs), and thymoglobulin, etc., dramatically reduced the incidence of severe acute TCMR. Furthermore, plasmapheresis and anti-CD20 Abs, which are recognized as desensitization, also brought better outcomes in ABO incompatible organ transplantation. Conversely, the issues regarding DSA are still disputable and seem not to be reasonably addressed. Therefore, discussions regarding ABMR and DSA are more frequently observed recently. This possibly reflects that the direction of researcher’s interests is shifting to ABMR due to DSA, especially for chronic types as our interests shift to long-term outcomes [1]. Nevertheless, it is also true that close correlations between TCMR and the onset of DSA inducing ABMR are becoming apparent. Thus, the effects of TCMR are being reassessed from perspective of long-range consequences. Therefore, the discussion only between DSA and ABMR seems to be simplistic and could be enriched by a broader view, which adds TCMR. In this section, we will review chronic ABMR, TCMR, and related topics in kidney, liver, and pancreas transplants.

### 3.1. Kidney Transplantation

Kidney transplantation was the first successful organ transplantation and is the most frequently performed. Therefore, a large amount of knowledge about rejection has been gained, and this has improved outcomes so far in this field. However, ABMR due to DSA remains a major barrier to achieving a good prognosis in kidney transplants. In recent years, the impact of preformed DSA on ABMR has become smaller, but the impact of newly produced DSA (de novo DSA (dnDSA)) on ABMR is still significant. dnDSA could be detected at the stage of performing a biopsy by observing mild renal dysfunction or at the stage of protocol biopsy performed for an asymptomatic recipient. Patients who produced dnDSA have been reported to demonstrate worse graft survival rates than recipients without dnDSA [1]. 

Given the fact that the consequences of dnDSA production are significant, it is reasonable to believe that prevention and early detection are crucial for adequate management. Monitoring tests for anti-HLA Abs with an immobilized single allele of purified HLA are available for the detection of dnDSA [2]. Recently, it has been reported that the graft immunocomplex capture fluorescent analysis (ICFA) method using transplanted tissue pieces obtained by allograft biopsies is also effective for the early detection of DSA production [3]. Pathological diagnosis is widely used as a gold standard in diagnosing ABMR based on the Banff Classification, which was initially reported in 1993 with the aim of creating an internationally unified standard for kidney transplant pathology [4]. The Banff classification has been updated every two years and is a standard that reflects the current situation and helps clinicians in making diagnoses. In addition to introducing chronic active ABMR in 2001 [5], the concept of chronic TCMR was partially introduced in the 2005 Banff Classification update, and the 2017 Banff classification update recently introduced the features of chronic active TCMR. Chronic active TCMR can be considered as a major factor resulting in interstitial fibrosis and tubular atrophy (IFTA). IFTA is also associated with poor long-term outcomes. Moreover, considering inflammation in IFTA areas may supply better outcomes [6].

It must be admitted that the development of dnDSA is closely related to T cell activities. Activated B-lymphocytes linage cells start to produce dnDSA as a consequence of the interaction between B cells and CD4 T cells through direct, indirect, and semidirect pathways [2]. Thus, TCMR theoretically appears to have an important role in subsequent ABMR, although limited evidence exists. In a recent prospective study, the reported incidence of TCMR in the first year after transplant was around twice as high as in recipients who developed dnDSA compared with patients without dnDSA. The presence of dnDSA was associated with the severity of TCMR and subsequent graft loss. Furthermore, patients with dnDSA, accompanied by more severe tubulointerstitial inflammation, were more prone to recurrent TCMR [7]. Another study demonstrated that recipients with dnDSA accompanied by a prior history of TCMR showed inferior graft outcomes [8]. Taken together, these studies suggest that dnDSA-related ABMR is preceded by T cell involvement and also that TCMR affects graft outcomes through ABMR.

The risks of dnDSA development may vary according to an individual’s immunological background. The rates of dnDSA development have been reported as being around 10% at 12 months after transplant under standard immunosuppressants [9,10]. However, recipients with high immunological risks (highly sensitized patients with high-titer of preformed DSA) are more susceptible to this particular development than those with low-risk, although preformed DSA became undetectable at 12 months after surgery with appropriate preparations [11]. In terms of immunosuppression, it is true, theoretically, that the levels of calcineurin inhibitors (CNI), mTOR inhibitors [12], antimetabolites [13], steroid, and several other monoclonal and polyclonal antibodies are related to the occurrence of dnDSA. Among them, CNI is one of the main immunosuppressants used to control dnDSA. High tacrolimus variability (instability) may well result in dnDSA production in pediatric recipients [14]. It can be admitted that this study has highlighted the importance of medication adherence. Furthermore, complete withdrawal from CNI by using mTOR inhibitors seems to favor dnDSA development [12,15]. Regarding steroid administration, this still plays an important role in controlling ABMR due to dnDSA [16]. On the other hand, there was a study that examined the effects of steroid withdrawal at 7 days post-transplant, and no negative effects on dnDSA were observed [17]. However, the latter study applied thymoglobulin induction for all recipients, which might alter immunological reactions. Therefore, the CNI and steroid-free strategies could involve immunological risks, although these approaches are attractive from the persepctive of medication side effects. It can be argued that the minimization of CNI by using mTOR inhibitors does prevent CNI toxicity. However, mTOR inhibitors without CNI or complete withdrawal from steroid administration pose a higher risk of DSA for recipients with standard induction therapy.

A variety of therapeutic approaches have been implemented, owing to the recent development of synthetic antibodies, in addition to orthodox therapeutic modalities. Several institutions applied steroid pulse, plasmapheresis, and intravenous immunoglobulin therapy (IVIG), combined with anti-CD 20 Abs administration for dnDSA mediated ABMR [18]. Other than these conventional therapies, several monoclonal Abs including eculizumab (anti-C5), tocilizumab, and clazakizumab (anti-interleukin 6) have recently been introduced in this field, although a proteasome inhibitor (bortezomib) failed to show therapeutic effects [19]. A pilot study to investigate the role of complement by using eculizumab showed a slight stabilization of renal function due to a terminal complement inhibitor, albeit with an underpowered design (treatment group *n* = 10) [20]. Choi et al. reported 80% graft survival at 6-year post-introduction in tocilizumab-treated patients for whom IVIG and rituximab, with or without plasma exchange, did not succeed [21]. Several other investigations of tocilizumab and clazakizumab followed and showed a significant reduction in dnDSA [22] and a suppression of eGFR decline, respectively [23]. However, these studies indicate that a certain population of patients did not show the efficacy of these treatments. Thus, patient selection for dnDSA treatment would be another key issue in this field. On the other hand, belatacept-based immunosuppression clearly prevented the development of dnDSA, compared with an ordinal cyclosporin-based treatment regimen [24]. These results remind us of the importance of suppressing dnDSA development. Furthermore, in addition to the introduction of novel medications, it is also pivotal to seek optimal regimens with existing therapeutic modalities for dnDSA mediated ABMR.

### 3.2. Liver Transplantation

Despite the significant effects on graft survival of ABO blood type-related ABMR, the role of dnDSA in liver transplant remains largely unknown. There are many studies that demonstrate serum dnDSA (s-DSA) and graft outcomes. These studies demonstrated that the existence of s-DSA is not necessarily related with poor outcomes [25], which is contrary to the facts in kidney transplant. In other words, it is not wise to assess ABMR of liver allografts only by s-DSA. In fact, clinically, the Banff working group advocates diagnostic criteria that consist of (1) pathological findings, (2) positivity of DSA, (3) C4d positivity, and (4) excluding other causative factors [26]. These studies, regarding dnDSA, possibly include s-DSA positive but intra-graft DSA (g-DSA) negative cases (i.e., the circumstance where allografts are free from damages due to s-DSA). It is important to distinguish these cases clinically, although limited availability of DSA examination hinders routine investigations of g-DSA. In fact, our recent study showed that g-DSA was closely associated with graft rejection and could be a more reliable indicator for the graft outcomes [27]. On the other hand, other research studies suggest that DSA clearance by Kupffer cells [28], release of HLA from liver in soluble form [29], allografts size [30], and liver regenerative abilities prevent harming liver allografts [31]. Thus, believing the information regarding s-DSA only without questioning may misconstrue the truth of what happens in liver allografts.

With the minimization of immunosuppressants, it is often observed that liver transplant recipients also developed dnDSA in serum, with the tendency of this belonging to class II rather than class I [25,32]. In terms of the relation between dnDSA and TCMR, the prevalence of dnDSA was clearly higher in recipients who experienced TCMR, albeit without a correlation with frequency and severity [33]. With regard to the effects of dnDSA development, the correlation between dnDSA and more severe fibrosis was observed [34,35]. These studies also demonstrated that rejection signs existed more frequently in the dnDSA positive group. Interestingly, s-DSA positivity in subclinical TCMR tended to be associated with more severe inflammation and fibrosis, while s-DSA negative subclinical TCMR resulted in no histological rejection, supported by gene transcriptional evidence [36]. In a relatively large study, the relation between dnDSA and fibrosis can also be observed by looking at 61 dnDSA positive patients among a total of 749 liver transplant recipients, albeit without pathological insights and HLA–C/DP information [37]. In this retrospective study, the most frequent target of HLA was the DQ locus (85%, 52 out of 61 patients). Cyclosporin usage rather than tacrolimus and low CNI levels (tacrolimus < 3 ng/mL, cyclosporin < 75 ng/mL in the first year) were detected as risk factors of developing dnDSA. On the other hand, high model for end-stage liver disease (MELD) scores and advanced recipient age (>60) functioned as protective factors. These findings appear to be consistent with immunological theories, i.e., deteriorating or elderly patients may exhibit milder immunological reactions.

Given the fact that the existence of dnDSA does not necessarily indicate the onset of ABMR, it is still unclear whether we should make interventions against the status of positive dnDSA. Conversely, it is better to seek remedies if ABMR is confirmed based on the Banff criteria. To address ABMR due to dnDSA, several therapeutic strategies have been applied so far. Simple reinforcement of immunosuppressants appears to be the first choice. Several cases showed a reduction in s-DSA MFI [32]. In addition, several institutions seem to apply other therapies for ABMR just as in renal transplant: anti-CD 20 Abs, IVIG, and plasmapheresis [38,39]. However, it seems to be rare to reverse the severe fibrotic changes after the onset of chronic ABMR. Thus, at present, it is of vital importance to prevent the development of dnDSA and subsequent ABMR.

### 3.3. Pancreas Transplantation

Due to improvements in surgical techniques and immunosuppression protocols, the graft survival of pancreas transplant has greatly improved. Similarly to other organs, with the pancreas, it is true that our attention has been shifting from these acute issues to ABMR. However, the association between dnDSA and graft outcomes is not well understood. Several studies have demonstrated the negative impact of ABMR on pancreas graft survival. However, many of these studies were limited by retrospective styles, small sample size, or lack of histological assessment. Among them, de Kort et al. initially investigated 27 pancreas transplant patients according to their s-DSA and biopsy findings. Graft survival largely depended on s-DSA positivity accompanied by complement activation (C4d positivity) [40]. Furthermore, they revealed that ABMR played a significant role in early graft loss < 1 year after transplant, especially in the setting of thrombotic cases [41]. Without histological assessment, a larger study consisting of 167 pancreas transplant recipients showed that about 15% (26/167) recipients developed s-DSA during a nine-year follow-up and resulted in significantly inferior outcomes: graft survival dropped around 30% and 20% from recipients without anti HLA Abs and DSA, respectively [42].

Regarding the class of DSA, a large part of dnDSA belonged to class II [43], especially DQ [44]. These results are also consistent with findings of kidney and liver transplants. Considering the recent developments in HLA matching, a more sophisticated approach should also become possible in the field of pancreas transplant. This approach demonstrated that the development of dnDSA after pancreas transplant was associated with the number of predicted indirectly recognizable HLA epitopes (PIRCHE) II [45]. Practically, it seems to be difficult to consider all information regarding HLA prior to deceased organ transplant. Nonetheless, it would be plausible to cater tailored immunosuppression based on their immunological status, provided that details of donors’ HLA information are supplied.

Although there is no research focusing on the optimal treatment for ABMR in pancreas transplant, the ABMR of pancreas grafts seems also to be managed by additional anti-humoral therapies with reinforcement of standard immunosuppressants [40,42,43]. However, it is controversial to initiate preemptive treatments for dnDSA positive cases without ABMR signs. Uva et al. added Belatacept to the maintenance of immunosuppressants for selected patients, although they denied universal preemptive interventions [46].

Overall, ABMR in pancreas transplant remains relatively unclear compared to other fields. Given the fact that many pancreas transplants have been conducted together with kidney transplants, research on ABMR in pancreas transplant would progress with the knowledge of kidney transplants. Indeed, there are criteria for ABMR of pancreas grafts [47]. Due to the hesitancy regarding pancreas graft biopsies, it must be admitted that an accurate assessment of rejection in pancreas graft is limited on several occasions. In addition to pancreas biopsies, there is a different approach: Duodenal graft biopsies have been taken [48]. This procedure may additionally shed light on pancreas graft ABMR according to intestinal transplantation and could improve outcomes. Limited knowledge of pancreas ABMR demands large prospective cohorts in the future.

Recent studies regarding chronic ABMR due to dnDSA in each field are summarized in Table 1.

## 4. An Overview of Recurrent Primary Disease

Recurrent primary disease plays an important role in determining graft outcomes in almost all fields of organ transplantation. Discussing the recurrence of primary disease always requires a clear definition of the recurrence. Described here are several reports regarding recurrences of primary diseases in the different fields. It is important, however, to keep in mind that misleading reports can exist due to an unclear definition of recurrence. Thus, in this review, the forms of recurrences are determined as follows for the sake of clarity: pathological recurrence (PR), mild-to-moderate clinical recurrence (mCR), and severe clinical recurrence (sCR) that results in end-stage organ failure.

### 4.1. Kidney Transplantation

There are many renal transplant recipients whose primary diseases are uncertain. In these cases, nephritis after transplantation cannot be recognized as recurrent or de novo nephritis precisely. However, recurrences of primary diseases certainly cause negative effects on graft survival rates. The management of recurrences should be considered seriously.

#### 4.1.1. IgA nephropathy

Recurrent IgA nephropathy (IgAN) is a crucial topic in renal transplant recipients with primary IgA nephropathy because a reported recurrence rate seems to be around 30% [52,53] at 10 years after transplant and as high as 50% in biopsied patients [54,55]. Although many of these recurrent incidents are characterized by only PR with a benign clinical course, around 10% of these cases over the course of 10 years were associated with the aggressive deterioration of renal function: sCR, resulting in graft loss. Thus, the control of IgAN recurrence remains an unmet need in the field of renal transplantation. Several factors may affect the pathogenesis of IgAN recurrence, such as low levels of immunosuppression, especially steroid avoidance [56] or HLA mismatch [57].

Cumulative data have suggested that circulating under-galactosylated or galactose-deficient (Gd) IgA1 and the subsequent generation of anti-GdIgA1 IgG (αIgA) play central roles in the pathogenesis of IgAN [58]. Suzuki et al. [59] reported that the levels of serum αIgA were closely associated with disease activity in native IgAN. As a progression of αIgA research, Julian and his colleagues [60] revealed glomerular deposition of αIgA in native IgAN by means of the extraction of biopsy specimens, although not having enough tissue prevented individual analysis. In terms of IgAN recurrence after renal transplantation, Berthelot et al. followed 60 IgAN transplant patients, concluding that high serum GdIgA1 resulted in IgAN recurrence. Julian’s group also demonstrated a similar result that serum normalized αIgA was an independent risk factor for IgAN recurrence [61]. In research described above, GdIgA1 and αIgA were measured by the conventional ELISA method. By using the ICFA method, we measured αIgA (serum/intragraft) and GdIgA1/αIgA immunocomplexes (intragraft) to investigate whether a causal relationship exists between αIgA and IgAN recurrence. In this report, the IgAN recurrence group demonstrated significantly higher serum αIgA levels at the time of recurrence confirmation. The IgAN recurrence group also exhibited higher intragraft αIgA and relatively higher ICs than those of a non-IgAN recurrence group [62]. According to these reports, therefore, it is reasonable to believe that GdIgA1 and αIgA are key molecules in IgAN recurrence and important targets to prevent recurrence. Nevertheless, there is no concrete evidence regarding therapeutics that target GdIgA1 and αIgA at this moment.

Several studies suggest strategies as a part of prophylaxis to suppress IgAN recurrence, although no established induction therapy exists. As a prevention of IgAN recurrence, the Jikei group adopted elective tonsillectomy 1 year after renal transplantation and examined the relationship between tonsillectomy and serum GdIgA1 and GdIgA1 deposition in tonsils and kidneys [63]. They revealed that elective tonsillectomy reduced the rates of IgAN recurrence, coupled with decreased immunoreactivities of GdIgA1. This report supports an interesting tonsil–kidney circulation based on GdIgA1.

Since HLA mismatch was suggested as a risk factor for the recurrence, there may exist a difference in outcomes between related or unrelated donor transplants [64]. Rodas et al. reported that full mismatches in HLA-B mitigated the recurrence of IgAN by examining 86 transplants, which included 38 living donor transplants [57].

Next, our concern is therapeutic modalities available in IgAN recurrence. Although there is no specific guideline for renal transplant recipients, many institutions basically follow native IgAN treatments consisting of steroid pulse [65], rituximab, tonsillectomy [63], and elimination of other exacerbating factors. A simple observation strategy often takes place for many PR cases, especially in elderly recipients, combined with lifestyle guidance. Either way, the current situation demands large prospective cohorts to establish a certain treatment strategy for IgAN recurrence.

#### 4.1.2. Focal Segmental Glomerulosclerosis

FSGS is one of the main indications for renal transplant in relatively younger populations and recurs with a variety of rates from 10 to 60% within a couple of years after transplantation [66,67]. Due to its humoral-type pathogenesis, immediate recurrence is common and can occur within 24 h after transplant. As a typical example of clinical recurrence (CR), massive proteinuria occurs immediately following reperfusion. At this initial point, fusion of the podocyte foot processes is the only finding by electron microscopic investigation. This progresses to a typical focal segmental sclerosis within a few months, which can be observed under an optical microscope. Under the current medical settings, a recurrence of FSGS significantly deteriorates graft survival and demonstrates around 30–60% survival rate at 5 to 10 years after transplant [68,69,70].

We would like to examine a typical example that suggests that pathogenesis of FSGS is based on a circulating factor [71]. An sCR-deteriorated allograft in a FSGS recipient can regain its function in a different non-FSGS recipient after re-transplantation. Furthermore, plasma obtained from FSGS patients can reproduce FSGS phenomena in rats [72]. Subsequently, intensive study has revealed that the soluble urokinase-type plasminogen activator receptor (suPAR) is involved in the onset of FSGS [73]. Podocyte effacement is also closely related to serum suPAR levels, while suPAR levels reflect the response to therapy [74]. Therefore, it is reasonable to believe that reduction in suPAR results in a low CR rate.

Considering the possibility of immediate deterioration, it is reasonable to carry out prophylaxis measurements prior to transplant. Several approaches have been attempted as induction therapies, and these approaches can be roughly divided into B-cell depletion and apheresis. Although there is a report of these approaches having no preventive effects [75], the effectiveness of plasma exchange or apheresis is advocated by many institutions. On the other hand, B-cell depletion is often accomplished by anti-CD20 Abs. Prior to transplant, B-cell depletion followed by a series of plasma exchange or apheresis is one of the induction therapies that can be used to mitigate recurrence. These effects could be explained by the reduction in circulating factors, including suPAR. As a donor selection, a living related donor may have disadvantages in terms of recurrence. This study demonstrated a 6.6% increase in recurrence rates at 10 years after renal transplant in a living related donor compared to an unrelated donor [76].

Current therapeutics for CR consist of reinforcement of immunosuppressants, including steroid pulse, rituximab, and plasma exchange [77]. Although suPAR levels rebound relatively quickly, it has been reported that CR status was stabilized following these therapies [78]. Further investigation into suPAR and other humoral factors may provide more sophisticated therapeutic approaches for FSGS.

#### 4.1.3. Membranoproliferative Glomerulonephritis

Membranoproliferative glomerulonephritis (MPGN) is evidenced by the thickening of capillary walls and diffuse mesangial cell proliferation under a light microscope. This disease can be divided into primary (unknown reason) and secondary (known reason) MPGN. Previously, MPGN had also been split into three distinct types based on pathological findings: sub-endothelial and mesangial immune deposits (type I, most common, mainly secondary MPGN); dense-deposit disease associated with the deposition of complement C3 without immunoglobulin deposition (type II); and a mixture of sub-epithelial and sub-endothelial immune deposits (type III, subtype of type I) [79]. Recently, this categorization has been replaced by a pathogenesis-based classification: (1) alternative complement pathway activation and (2) immunoglobulin-related type. C3 glomerulopathy belongs to classification (1) and is determined where glomerulonephritis is accompanied only by C3 deposition without C1q or C4 deposition. C3 glomerulopathy has two forms: dense-deposit diseases and another C3-deposited glomerulopathy, which were previously categorized in type I and III [80,81].

Recurrence rates at 11.8%, 15.6%, and 18.9% at 5, 10, and 15 years, respectively, after renal transplantation have been reported [70]. More prominently, C3 glomerulopathy exhibits higher than 80% recurrence rates in a small case series (19 patients) [82]. About 50 to 70% high rates of graft failure within 5 years after recurrence were observed in MPGN recurrent cases, especially in dense-deposit disease [70,82]. Thus, it is pivotal to explain these serious outcomes after recurrence for renal transplant candidates in advance. Furthermore, we should seek circumstances where recurrence is less likely to occur. An Israeli group reported that 19% recipients of MPGN type I recurred over a 118 ± 61 months follow-up period, and HLA B49 and DR4 were considered to be risk alleles. This research also suggests that an unrelated donor is preferred for transplant due to MPGN, based on the finding that higher rates (25%) of living related to donor renal transplant recipients recurred MPGN rather than to unrelated donor transplant (0%), although another study denied the participation of donor category in recurrent MPGN [76]. In addition, recurrent MPGN, again, clearly showed worse graft survival [83]. Although an optimal induction therapy has not been determined, a B-cell targeted immunosuppressive approach appears to be logical.

The strategies for recurrence of MPGN are also similar to those for the primary disease, which consist of plasma exchange, rituximab, steroid pulse, or eculizumab [84]. Nevertheless, given the significant rates of recurrence and graft failure, it is crucial to inform patients of the outcomes of renal transplantation and to establish effective therapeutic strategies.

#### 4.1.4. Membranous Nephropathy

Membranous nephropathy (MN) occurs as a pure glomerular-specific autoimmune disease, as well as a secondary disease due to a systemic condition such as infections, malignant disorders, autoimmune diseases, etc. This entity is induced by immunocomplex deposition in the sub-epithelial area of the glomerular basement membrane [85]. The primary cause seems to be a development of Abs against podocytes. The major Abs that have been identified include anti-phospholipase A2 receptor antibodies (PLA2R) [86] and thrombospondin type-1 domain-containing 7A antibodies (THSD7A) [87].

Recurrence rates at 10% (5-year), 16% (10-year), and 18% (15-year) were observed, which are similar to those of MPGN discussed above. The recurrence rates could be higher (up to 40%) when including PR found by a protocol biopsy [88]. PR is likely to occur within 1 year after transplant and has a possibility of recurring as early as 2 weeks post-transplant. The secondary MN displayed relatively low rates of recurrence, provided that the primary diseases were well controlled. The recurring MN could result in notable graft failure rates of up to 60% [70]. A lack of consistency in the graft failure rates in recurring cases would indicate that the studied population was not heterogenous.

Regarding recurrence and autoantibody levels, Kattal et al. investigated 26 MN recipients according to their anti-PLA2R levels and revealed that anti-PLA2R levels were 83% and 42% of the positive and negative predictive values, respectively, for recurrent MN after transplant [89]. In addition, anti-THSD7A also could induced a recurrence of MN, as evidenced in a case report accompanying an investigation into a murine MN model [90]. Therefore, autoantibody levels may enable us to stratify recipients into appropriate recurrent risk groups.

In order to prevent recurrence, many researchers may seek to perform treatments with induction therapy prior to transplant. However, at present, there appears to be no reliable induction therapy for meaningfully suppressing recurrence. Furthermore, it is still controversial whether unrelated donors are advantageous to recurrent MN [76,91].

The management of recurring MN, again, follows primary MN treatments consisting of treatments for nephrotic syndrome and decreased renal function.

Features of IgAN, FSGS, MPGN, and MN with overall recurrence rates are described in Table 2.

#### 4.1.5. Lupus Nephritis

Generally, lupus nephritis is diagnosed in various rates up to 70% of patients with systemic lupus erythematosus (SLE) [95]. Typically, the onset of nephritis is observed within 3 to 5 years after SLE diagnosis [96]. Pathologically, nuclear antigens, especially DNA and anti–nuclear/DNA complement-binding IgG, seem to play important roles in the onset of lupus nephritis [97].

CR after renal transplantation can be less than 5% [98]. It can be argued that pathological activity decreases at the time of renal failure and the introduction of immunosuppression after transplant also suppresses disease activity. A large study for lupus nephritis indicated equivalent outcomes relative to non-lupus nephritis renal transplant recipients [99]. Deegens et al. reported that one patient out of 23 recipients exhibited lupus nephritis, albeit with no biopsy evidence due to coagulopathy [100]. However, it is true that the recurrence rate largely relies on the executing rates of biopsy procedures. In fact, PR increased by up to 50% when less aggressive types of histological changes—class II, III, etc.—are included [101]. Taken together, PR seems to be relatively common, but does not affect overall outcomes. Therefore, it is reasonable to perform renal transplant for lupus-related renal failure.

Regarding induction therapy, there is no standard recommendation at present. However, it is recommended that renal transplant should be performed after the introduction of dialysis therapy for several months, with the dosage of prednisolone decreased to less than 10 mg/day [102]. From this perspective, preemptive renal transplant may not be feasible, especially in cases with rapid progression to renal failure. When considering preemptive transplant, disease activities assessed by anti-nuclear Abs and anti-double strand DNA Abs, CH50, C3, etc., should be carefully reviewed.

For CR cases, treatment plans consist of steroid pulse, increasing the dose of mycophenolate mofetil or substituting cyclophosphamide for mycophenolate mofetil, and anti-CD20 Abs [103].

#### 4.1.6. Anti-Neutrophil Cytoplasmic Autoantibody or Anti-Glomerular Basement Membrane Antibody Positive Rapidly Progressive Glomerulonephritis

Both anti-neutrophil cytoplasmic autoantibodies (ANCA) and anti-glomerular basement membrane (GBM) antibody-positive rapidly progressive glomerulonephritis (RPGN) are categorized as acute progressions to renal failure accompanied with hematuria, proteinuria, and anemia. Clinical RPGN comprises a variety of diseases such as part of IgAN, thrombotic microangiopathy, acute interstitial nephritis, etc. This section features ANCA-related nephritis and anti-GBM-Abs-related nephritis [104].

Around 10% recurrence rates have been reported for ANCA-related nephritis after transplant, and one-third of these resulted in graft loss over the first five years [105]. Although ANCA titers are utilized to assess disease activity, several studies indicate that the levels of ANCA could not be relied upon to assess recurrence [106,107]. Nonetheless, the activity of primary disease is considered important for controlling recurrence. Thus, it is sensible to wait for renal transplant at least one year after the activity becomes under control [108].

On the other hand, the recent recurrence rates for anti-GBM-Abs-related nephritis seem to be less than 5%, which is lower than that of ANCA-related nephritis [109,110]. Interestingly, the step for waiting for remission is similar, but a decrease in anti-GBM Abs for at least 12 months consecutive is also required for safe renal transplant [111].

It is feasible to perform renal transplants for these two RPGN-induced renal failures when indicated due to the relative lower rates of recurrence.

Clinically important information regarding lupus nephritis, anti-ANCA, and anti-GBM RPGN is summarized in Table 3.

#### 4.1.7. Amyloidosis and Mimickers

Amyloidosis is a systemic disease and a relatively rare entity for renal transplantation. Amyloid deposition is observed under the electron microscopy at around 5–12 nm fibrils, which are β-pleated sheets [114]. In addition, a positive Congo red stain and an apple-green birefringence with polarized light are often used for confirmation [115]. Mass spectrometry became a useful tool for distinguishing the following subtypes [116].

AL (fibrils due to immunoglobulin light chain clonal production); AA (serum amyloid A congregation, secondary amyloidosis); ATTR (hereditary amyloidosis, genetic mutation of misfolding-prone protein (mainly transthyretin)); and ATTRwt (wild-type transthyretin misfolding). Responsible proteins of ATTR include apolipoprotein A-I, A-II, lysozyme, fibrinogen, and cystatin C, etc., in addition to transthyretin [117,118].

Although amyloidosis negatively impacted the patients’ graft survival, the overall outcomes were equivalent with diabetes mellitus (DM) or elderly (>65 years) recipients after renal transplantation. Around 15% recurrence rates had been reported, depending on the subtype of amyloidosis, and significantly affected patients’ survival. Regarding AL amyloidosis, a complete hematologic response seems to be key in achieving better survival [119] and acceptable outcomes (median duration to graft loss: 10.4 years), whilst patients with partial or no response demonstrated inferior outcomes (5.5 years) [120]. Apolipoprotein A-I and lysozyme amyloidosis showed better graft survival, with 13.1 years at median [119]. These studies suggest that renal transplant outcomes largely rely on the type of primary amyloidosis. Therefore, at least, it is reasonable to evaluate patients carefully and manage the primary amyloidosis appropriately in order to mitigate the risk of recurrence when considering renal transplant for amyloidosis.

On the other hand, as a mimicker, fibrillary glomerulonephritis (FGN) shares several common characteristics both clinically and histologically: nephrotic or nephrotic syndrome, deposits of 12–24 nm fibrils, and occasionally positive Congo red stain (4%). FGN is characterized by DnaJ homolog subfamily B member 9 as an auto-antigen [121] and possibly recurs after transplant with the rate at around 20% [122]. Thus, it is pivotal to make a correct diagnosis in confusing cases by using mass spectrometry.

In addition to recipients diagnosed with amyloidosis, it is essential to keep renal amyloidosis in mind in cases of new-onset proteinuria or nephrotic syndrome following renal transplantation for an unknown primary disease, since a routine pathological screening may not identify the early signs of recurrent amyloidosis.

### 4.2. Liver Transplantation

After liver transplantation, a variety of medical problems might occur, such as renal damage, new onset of DM, etc., in addition to rejection and infection. Furthermore, primary disease recurrences are another significant issue. There are at least three major mechanisms by which a primary disease recurs in liver grafts. The first category is a recurrence in the same manner as the primary disease, for example, primary biliary cholangitis or primary sclerosing cholangitis. Secondly, a recurrence of hepatitis due to a virus that is responsible for the primary hepatitis belongs to this category. Third, this category contains recurrences of tumors in liver graft. It can be recognized as primary tumor metastases to liver grafts.

#### 4.2.1. Viral Hepatitis

Although viral hepatitis B and C have been the leading reasons for adult liver transplants [123], a recent rapid increase in nonalcoholic steatohepatitis has rearranged this trend [124].

Regarding hepatitis B virus (HBV) reactivation following transplant, there are two main distinctive situations. The first instance is HBV hepatitis in HBsAg (hepatitis B surface antigen)-positive recipients, whereas HBV transfer from a donor who is HBsAg-negative/hepatitis B core antibodies (anti-HBc Abs)-positive is another example. From the perspective of recurrence, former conditions will be discussed here.

For recipients with HBV infection, inappropriate prophylaxis results in HBV reinfection of liver grafts. Following the removal of the infected liver, the reinfection is established with HBV remaining in the recipient’s blood stream. This trend seems to be more apparent in recipients with preoperative HBV-DNA positive than HBV-DNA and hepatitis B e antigen (HBeAg) double-negative cases. Eighty-three percent of preoperative HBV-DNA positive recipients had serums that became HBsAg-positive after surgery. Nevertheless, HBV-DNA and HBeAg double-negative cases also showed an HBsAg resurgence in 58% of recipients [125]. Therefore, HBV exists in recipients, albeit with negative results for HBV-DNA. It is well known that the outcomes of HBV reinfection have significant negative effects on liver grafts: rapid progression to cirrhosis [126]. As prophylaxis strategies, initially, hepatitis B immunoglobulin or lamivudine administration had been implemented. However, higher than 30% reactivation was confirmed despite prophylaxis [125,127]. Thereafter, it was proved that hepatitis B immunoglobulin and lamivudine coadministration was effective enough to suppress reactivation, with a rate of 0–10% [128,129]. Consequently, it became a standard practice to apply preoperative lamivudine treatment, intraoperative hepatitis B immunoglobulin IV, and postoperative coadministration.

Although hepatitis C virus (HCV) has spread worldwide, the introduction of direct-acting antivirals (DAA) has completely changed its management after liver transplantation [130]. In addition, DAA provides an opportunity to expand a donor pool to the HCV-positive population [131]. As a natural course of liver transplant for HCV, the rates of HCV-recurrent infection are significantly high and typically occur soon after surgery, while only 5% of recipients may escape from recurrence. In many cases, HCV RNA becomes detectable two to four weeks after transplant. Subsequently, many recipients show histological chronic hepatitis [132,133]. Before the introduction of DAA, it was believed that these factors might affect the overall outcomes of liver transplants. Surprisingly, these negative influences may be limited to slight-to-moderate inferior outcomes in 5-year patient survival rates [134]. Conversely, several studies have reported that HCV recurrence possibly caused serious outcomes, such as fibrosing cholestatic hepatitis [135], immediate progression to cirrhosis [136], or deteriorated patient survival [137].

Before the emergence of DAA, several treatment strategies were reported. From the 1990s, interferon therapy commenced [138]. Ribavirin was added in the 2000s [139] and was later improved by Peg-interferon [140]. However, the sustained virological response (SVR) after transplant remained only around 10–40%, which was far below acceptable rates [141,142]. In 2011, telaprevir was approved as a first-generation proteinase inhibitor and established as a triple-drug treatment [143]. The second generation simeprevir, emerged in 2013, and the regimen further improved [144].

Next, in 2014, the interferon-free DAAs, asunaprevir, and daclatasvir were developed with successfully high SVRs [145]. Conversely, it is true that candidates for liver transplant may have belonged to a category where preoperative treatment was ineffective or impossible due to refractory mutated HCV or patients’ other conditions. Thus, SVRs after transplant were inevitably lower than those of native HCV patients [146].

Considering these serious consequences, the effective DAA introduction was significant, especially after the emergence of sofosbuvir and ledipasvir. As the SVRs had reached nearly 100% both in native HCV and liver transplant patients [147,148], DAA introduction after liver transplant became a standard therapy. Furthermore, for recipients with renal failure, cirrhosis, or prior DAA failure, the effectiveness of glecaprevir and pibrentasvir has been reported [149].

#### 4.2.2. Malignant Tumor

Liver transplantation is indicated for end-stage liver disease with tumors or unresectable malignant liver tumors under certain conditions. Hepatocellular carcinoma (HCC) is a representative tumor that is most frequently indicated for liver transplantation. Originally, it is well known that the outcomes of liver transplantation for HCC without staging were inferior to those of other primary diseases without HCC [150]. This is primarily because HCC recurs in transplanted liver grafts in immunocompromised patients. Thus, liver transplantation should proceed in circumstances where HCC is not disseminated to the outside of the liver. To overcome the recurrence of HCC, the Milan criteria was clinically introduced worldwide [151]. However, a certain patient group demonstrated the equivalent outcomes, albeit with deviation from the Milan criteria. The criteria were then modestly expanded to the University of California, San Francisco (UCSF) criteria, which showed a 75% survival rate in 5 years [152]. The Kyoto group also expanded the eligibility for liver transplantation with an excellent survival rate of 82% and a low recurrence rate of 7% at 5-years post-transplant [153]. It is noteworthy that the Kyoto group not only applied tumor size and numbers, but also serum des-gamma-carboxy prothrombin levels as an assessment for tumor activities. Recently, there have been further efforts to establish several other safer models, such as the hazard associated with liver transplantation for hepatocellular carcinoma which incorporates a dynamic α–fetoprotein response [154].

Early stage unresectable cholangiocarcinoma is also considered as an indication for liver transplantation. Cholangiocarcinoma is the second most common liver cancer and pathophysiologically divided into two different groups: extrahepatic cholangiocarcinoma and intrahepatic cholangiocarcinoma. Among extrahepatic cholangiocarcinoma, perihilar cholangiocarcinoma may be an indication for liver transplantation, whilst liver transplantation is not indicated for distal cholangiocarcinoma given its location [155]. LT can only be a valid treatment plan if it can promise a better survival rate compared to liver resection. In order to achieve this, the Mayo Clinic protocol that was originally reported in 2000 has been supported. In summary, liver transplantation is performed following neoadjuvant chemoradiotherapy, which consists of extra-beam and transcatheter radiation therapy and intravenous 5-FU administration [156]. Based on experience, elevated CA19-9, encased portal vein, and non-radical resection were identified as predictors of recurrence following liver transplantation [157]. On the other hand, several initial studies regarding intrahepatic cholangiocarcinoma seem to be difficult to interpret since the pathological conditions of recipients were not sufficient. However, recent studies show that liver transplantation for very early intrahepatic cholangiocarcinoma—a single tumor measuring less than 2 cm—promises acceptable outcomes, with a 73% 5-year patient survival rate [158]. In summary, these results were obtained from incidental pathological findings in recipients with cirrhosis or initial misinterpretation as HCC. Thus, initial diagnosis such as very early intrahepatic cholangiocarcinoma should first be considered for liver resection. However, if liver resection is not an option, as with portal hypertension, etc., liver transplantation might be the last resort.

#### 4.2.3. Primary Biliary Cholangitis

Primary biliary cholangitis (PBC) is categorized as a cholestatic liver disease accompanied by autoimmune features. These features comprise anti-mitochondrial Abs (AMA) positivity (>90% of patients), targeting anti-E2 domain of pyruvate dehydrogenase complex Abs [159], and meaningful overlap with other autoimmune disorders, such as Sjogren’s syndrome and thyroiditis, etc. [160]. Thus, PBC is recognized as part of a systemic autoimmune condition. Histologically, PBC is characterized by a chronic destructive form of nonsuppurative granulomatous lesions with or without lymphocytes-infiltrate cholangitis in small-sized and medium-sized biliary trees [161].

With the introduction of ursodeoxycholic acid (UDCA) in the early stages of PBC, the resulting rates of liver transplant or death improved to 6% and 22% at 10 and 20 years, respectively, after the onset of PBC [162]. The outcomes of liver transplant are generally good, with around 80% and 70% survival rates reported at 5 and 10 years, respectively [163,164]. Interestingly, similar outcomes have been reported with deceased and living liver transplant donors [165]. The rates of recurrent PBC appear to vary depending on whether recipients received prophylactic UDCA administration and the frequency of liver biopsies. Recurrence rates at 10% to 20% have been reported in recipients with prophylaxis during an approximately 10-year follow-up period [166,167], whereas patients without prophylaxis have demonstrated greater than 30% recurrence rates over the same time period [168,169]. These studies support that UDCA is effective for the majority of patients with recurrence, but there were no significant improvements in histological changes and patients’ survival [169].

Diagnosis of recurrent PBC is largely reliant on histological findings, since only around 10% of recurrent PBC patients demonstrate classic symptoms of PBC, i.e., pruritis, jaundice, xerostomia, or keratoconjunctivitis sicca, etc. In order to standardize recurrent PBC, pathological features combined with the existence of AMA and elevated IgM are often adopted clinically. Pathological features include the following four findings: 1. epithelioid granulomas called florid lesions, 2. lymphoplasmacytic infiltration, 3. lymphocytes aggregation, and 4. bile duct injury. Definite recurrent PBC is defined by having all 3–4 pathological features, while conditions meeting 2/4 criteria are considered as probable recurrent PBC [170].

Citing definite risk factors for recurrence of PBC is still controversial, although several studies suggest that immunosuppressants, HLA alleles, HLA mismatch, recipient/donor age, and gender play a role, which appear to be feasible from the perspective of the behavior of autoimmune nature. Manousou et al. reported that therapy with cyclosporin and azathioprine in combination had preventive effects on the recurrence of PBC, although cyclosporin and tacrolimus alone had minimal influence [168]. Egawa et al. also suggest that there is superiority in conversion to cyclosporin from tacrolimus within 1 year to decrease the risk of recurrence, albeit with a disadvantage of cyclosporin as a primary calcineurin inhibitor [171]. This research enumerated several other risk factors: serum IgM 554 mg/dL or higher and donor-recipient sex mismatch. Recent meta-analysis from six retrospective studies showed tacrolimus inferiority and UDCA protective effects for the recurrence of PBC [172].

Regarding disease activity of recurrent PBC, it basically exhibits an indolent style that barely requires re-transplantation and tends not to affect long-term outcomes. A Japanese multicenter study showed that PBC recurrence rarely became an indication for re-transplantation after analyzing seven re-transplant cases from 516 liver transplant recipients of PBC [173], while Charatcharoenwitthaya et al. reported that two out of thirty-eight recurrent cases required re-transplantation [169]. Taken together, it can be argued that PBC recurrence seldom results in graft loss.

#### 4.2.4. Primary Sclerosing Cholangitis

Primary sclerosing cholangitis (PSC) is characterized by both intra-hepatic and extra-hepatic multiple or diffuse bile duct stenoses, which result in a chronic cholestatic conditions associated with cirrhosis. Pathogenesis of PSC remains unclear, but, as with other multifactorial diseases, genetic factors along with environmental factors may play a role in its onset. Notable findings include a strong association of up to 80% with inflammatory bowel disease, especially ulcerative colitis [174]. In addition, PSC increases the risk of primary liver cancer, especially cholangiocarcinoma, by up to 1500 times compared with the general population and increases the annual morbidity rate by 0.5% [175]. At present, liver transplantation is the only established therapy for PSC. The overall survival rates without liver transplant in PSC patients were 78 and 60% at 10 and 20 years, respectively, following the onset of PSC [176].

The outcomes of liver transplantation for PSC differ from relatively poor to acceptable graft survival rates. From the European Liver Transplant Registry, 80, 69, and 57% graft survival rates at 1, 5, and 10 years, respectively, were reported [163], while U.S. data showed 86.5, 78, and 71.5% for the same time periods [164]. Although several discrepancies exist, the outcomes are generally worse than those for liver transplants for other cholestatic diseases. It can be argued that recurrent PSC in transplanted livers is one of the reasons for the worse outcomes.

Periductal concentric fibrosis (onion skin fibrosis) accompanied with lymphocyte infiltration is a well-known histological finding. However, this typical feature is not often observed in PSC clinically. In order to diagnose PSC, cholangiography is considered the most important step. A beaded or “pruned tree” appearance and band-like strictures are characteristic features of PSC. Recurrent PSC is determined in PSC recipients who demonstrate these cholangiography findings or pathologies, with the exception of cases of hepatic artery thrombosis or other similar conditions [177]. Pathological features are considered particularly important in the diagnosis. Recurrence rates of around 20% have been reported, especially when less than a median of 5 years has passed since transplant [178,179,180].

The discrepancies in the reported incidences of recurrence may reveal some risk factors, although several risk factors have been reported from various institutions. These risk factors are considered to be possible issues for grafts, recipients–donor relations and recipients. Regarding liver grafts, marginal or extended donor criteria grafts may exhibit a higher incidence of recurrence [179]. Meaningful insights can also be obtained from living donor liver transplants. Notably, related (parent/child) pairs may show higher recurrence rates, and this tendency is clearer if recipients are followed-up longer: The hazard ratio is 3.12 (>12 months follow-up) [181]. Another study suggested higher recurrence rates in living donor liver transplants, compared to deceased donor transplant [182]. However, a multi-center cohort study denied the impact of donor type, albeit without collecting adequate HLA data [183].

In relation to recipient issues, several factors have been discussed, such as inflammatory bowel disease, rejection/immunosuppressants, younger age, and high MELD scores, etc. Of these, topics regarding inflammatory bowel disease are often discussed. Given the “leaky gut” hypothesis, unchanged bowel bacterial flora or substances from those after transplant may evoke the same pathological condition in transplanted liver grafts [184]. Thus, pretransplant colectomy with a remission of inflammatory bowel disease may confer PSC pathogenesis-free circumstances. Rejection, especially a refractory one, also imposes a higher risk of recurrence that may be explained by the fact that PSC and rejection share the same immunological pathways [185]. Taken together, several institutions seem to apply the following as clinically acceptable strategies: (1) avoidance of a first-degree relative as a donor and reliance on a deceased donor, if trusting the results that showed living donor inferiority, and (2) minimal withdrawal of immunosuppression.

Recurrent PSC tends to demonstrate CR style and demonstrates inferior outcomes (graft loss) that require re-transplantation with high probabilities (30–70%) compared to recurrent PBC or autoimmune hepatitis [181,183]. These results suggest that recurrent PSC progresses vehemently to graft loss. As with native PSC, no standard therapy has yet been established for recurrent PSC except for re-transplantation, although biliary tract drainage can be attempted in several cases as rescue therapy.

#### 4.2.5. Autoimmune Hepatitis

Autoimmune hepatitis (AIH) is an inflammatory, basically progressive, autoimmune disease that mainly affects middle-aged women (but possibly all ages and both genders) due to uncertain causes. The disease progression seems to rely on environmental triggers and genetic factors [186]. Although no disease-specific markers have been identified, several autoantibodies have been recognized in AIH: anti-nuclear, anti-smooth muscle, anti-liver kidney microsome, and anti-liver cytosol antigen type I. Based on these serological markers, AIH can be separated into type I (anti-nuclear or anti-smooth muscle Abs, or both) and II (anti-liver kidney microsome Abs) AIH [187]. In addition, the existence of type III AIH (anti-soluble liver antigens Abs) has been advocated [188], although the characteristic of this entity overlaps with type I AIH [187]. AIH is also characterized by mildly elevated serum IgG levels and interface hepatitis or plasma-lymphocytic infiltration. In terms of treatment, the first recommendation is corticosteroids followed by azathioprine and mycophenolate mofetil in that order [189]. If appropriate therapies are provided, the long-term outcome of AIH generally provides sufficient life expectancy. However, compared to other chronic liver diseases, the lack of these interventions results in relatively rapid progression to cirrhosis and liver failure, which requires liver transplant [190].

The recurrence of AIH after liver transplantation was initially reported with the rate of 26%, including PR to CR, by the Pittsburgh group [191]. Several other studies reported similar results of around 30 to 40% recurrence rates [192,193], including ambiguous cases. According to a recent systematic review, 8–12% and 36–68% PR to CR have been reported in one and five years, respectively, after liver transplant [194]. PR, with an indolent clinical course, may make up a large part of the recurrence. Although there is no firm consensus regarding recurrence, the diagnosis of recurrence largely relies on clinical manifestations: abnormal liver function tests, positive autoantibody status, high gamma globulinemia, and pathological findings (lymphoplasmacytic infiltration to portal area, central perivenulitis, interface hepatitis, and foci of necrosis) without evidence of endothelialitis and ductulitis, with the exclusion of rejection and viral infection. Notably, it is important to pay attention to the existence of TCMR, because the frequency of TCMR is generally higher in AIH, which may influence the diagnosis of recurrence [195]. 

Genetic factors may also play roles in the recurrence of AIH. Several studies over the last two decades detected HLA-DR isotype involvement. Two initial reports suggested that HLA-DR3 positivity in recipients has a negative impact on recurrence [192], especially in HLA–DR3 negative allografts [191]. A recent study also described how mismatches on both HLA-DR alleles results in a significant risk of recurrence, especially for patients with single-agent immunosuppression. This study also showed that racial factors might play a role in developing a recurrence [196]. It is also true that the high activity of primary AIH before transplant influences the rates of recurrence. In a multivariate analysis, the degree of inflammatory activity and high IgG levels were recognized as risk factors for recurrence [197]. Information regarding recurrence rates based on the type of AIH seems to be limited [198].

Treatment strategies for recurrence are based mainly on primary AIH and consist of steroid, azathioprine, and mycophenolate mofetil. In recipients with risk factors, a certain level of immunosuppression is required, although a steroid-minimization approach is often applied in the liver transplant field. The long-term outcome of recurrence may show slight disadvantages in graft survival rates [199], but there appears to be no significant difference between non-recurrence and recurrence populations [196].

These three autoimmune liver diseases are summarized in Table 4 from the perspective of liver transplantation.

### 4.3. Pancreas Transplantation

Although the target diseases for pancreas transplantation may have a narrow range, recurrent primary DM could have a large negative impact on graft survival. Pancreas transplant is basically indicated for patients with type 1 insulin dependent DM or after total pancreatectomy. However, several studies, discussed below, showed that type 2 DM patients without significant insulin resistant can also become candidates for pancreas transplant with acceptable outcomes. In order to control primary outcomes, DM is a key for achieving excellent outcomes. In this section, we describe type 1 DM, which may recur after transplant.

#### Type 1 Diabetes Mellitus

Type 1 DM can be classified as an autoimmune disease. Pancreas-islet-related autoantibodies are frequently identified. Although up to 20 or more different autoantibodies have been reported, islet cell Abs (ICA), insulin autoantibody (IAA), glutamic acid decarboxylase 65 (GAD65) antibody, insulinoma-associated protein-2 (IA-2) antibody, and zinc transporter 8 (ZnT8) autoantibody have been considered as clinically important autoantibodies [204]. ICA was originally discovered by Bottazzo et al. in 1974 [205]. Measuring ICA is the gold-standard method for diagnosing and predicting type 1 DM, primarily due to its high sensitivity and specificity. However, it is unreasonable to use this in the clinical setting, as it involves a complicated procedure. Furthermore, GAD65 and IA-2 have been identified as the mainly corresponding antigens of ICA [206]. Therefore, alternatively, anti-GAD65 and IA-2 Abs seem to be substituted clinically for ICA.

The overall rate of developing type 1 DM after pancreas transplantation is relatively low, provided that effective immunosuppression is introduced [207]. However, as a typical example, recurrence occurred in twins or HLA-identical siblings who underwent living-related pancreas transplantation with minimal effects of immunosuppression [208]. These studies point to the importance of common HLA sharing and immunosuppression. The existence of HLA-DR3 and HLA-DR4 in the recipient’s allele is particularly considered as a risk factor. HLA-DR allele sharing also seems to be an unfavorable factor regarding recurrence [209]. Conversely, even though conventional immunosuppression was introduced for HLA-mismatched pancreas transplantation, type 1 DM recurrence was observed. It has been reported that autoantibody positivity is related to poor glucose tolerance, although this study did not include histologic examination and discards the possibilities of other causes [210].

Thus, it is reasonable to believe that a certain population is vulnerable in nature to the recurrence of an autoimmune disease. By employing a progression of immunosuppressive medications, the Miami group reported a reduction in the recurrence rate in type 1 DM. This effect may be due to induction therapy, such as anti-CD25 antibody or thymoglobulin, rather than maintenance immunosuppression [209]. However, the administration of 3-4 medications in combination may additionally suppress recurrence.

## 5. Feature Perspective and Concluding Remarks

ABMR and a recurrence of primary disease are looming subjects that hinder the achievement of excellent long-term graft survival. It has been considered that ABMR is prominent in renal transplant rather than liver and pancreas transplants. Nevertheless, as a development of immunological assessments, the identification of ABMR in liver and pancreas transplants has become more apparent than before. Due to the fact that there seems to be common immunological reactions in these different organs, integrated approaches from different fields could accelerate the understanding of ABMR. Furthermore, knowing the nature of the primary disease would also facilitate tailoring immunosuppression to mitigate the risk of recurrence.

Lowering immunosuppression under a certain threshold could trigger the onset of these two distinctive conditions. Thus, it would be ideal to find more precise thresholds for each case by monitoring the recipients’ immunological status. Regarding induction therapies, a single induction at pre-transplant or peri-transplant may be unfeasible with respect to maintaining the condition free from both ABMR and recurrence in the long term, because the recovery of immunity occurs over the course of post-transplant. However, more specific tolerance induction or complete remission from the primary disease could alleviate the risk of both entities or realize a better condition. ABMR and recurrence of primary diseases have their own preferences regarding donor immunological backgrounds, which would be contrary to each other. Donor selection could be arranged by estimating the advantages and disadvantages. Adjustment of these factors could result in improved outcomes in organ transplantation.

## Figures and Tables

**Table 1 jcm-10-05417-t001:** Summary of recent studies in chronic ABMR due to dnDSA.

Renal Transplant	Study Design, Number of ABMR Patients (Number of Treated Patients)	Treatment	Median (or Mean) Post-Transplant/Post-Treatment Follow-Up Period (Months)	Major Outcomes	Reference
	RCT, 25(12)	Rituximab + IVIG vs. placebo	(118)/12	Treatment group eGFR decline was slightly smaller but not significant.	Moreso 2018 [49]
	RCT, 20(10)	Eculizumab vs. control	N/A/6	Slight stabilization was noted in renal function while on treatment.	Kulkarni 2017 [20]
	Prospective, 36(36)	Tocilizumab	N/A/39.1	Significant reduction in DSA and stabilization of renal function.	Choi 2017 [21]
	RCT, 20	Clazakizumab vs. placebo	10.6 (4.4–16.2) years post-op to inclusion in the trial/52 weeks	The mean eGFR decline during treatment was notably slower.	Doberer 2020 [23]
	RCT, 44(21)	Bortezomib vs. placebo	N/A/24	No significant improvement in GFR decline in the treatment group despite significant toxicity.	Eskandary 2018 [19]
	Retrospective, 123 (108 at least steroid + IVIG)	Steroid pulse+IVIG(+rituximab, PP, thymoglobulin)	9.5 (2.7–20.3)/4.3 (0–8.8) from diagnosis of ABMR	The combination of steroid pulse and IVIG demonstrated a reduced risk of graft loss.	Redfield 2016 [50]
	RCT, all patients: 660 (219/226/215)	Belatacept more intense vs. Belatacept less intense vs. cyclosporine treated	Up to 7 years follow-up from randomization	Belatacept-based immunosuppression effectively suppressed DSA production.	Bray 2018 [24]
Liver Transplant					
	Retrospective, 9(9) (dnDSA + acute ABMR patients)	Steroid, IVIG, PP, rituximab, ATG, retransplantation	44(13–66)/36 (3–65)	Seven out of nine recipients demonstrated stable liver enzyme tests.	Del Bello 2015 [38]
	Retrospective, 4(3 pediatric)	IVIG	N/A/N/A	IVIG had minimal effects on MFI of DSA.	Guerra 2017 [51]
	Retrospective, 9(9, pediatric 4, adult 5)	Rituximab + α(IVIG, Bortezomib)	104(17–245)/60(5–65)	The administration of rituximab for chronic ABMR may be feasible.	Sakamoto 2021 [39]
PancreasTransplant					
	* Retrospective, various treatments, 9(4)	(nonstandard treatment for 4 patients) ATG, IVIG, PP, alemtuzumab, pancreatectomy	21.7(range 0.1–169.5) months/N/A	Three patients received pharmacological treatments and 4 out of 9 patients lost their graft.	de Kort 2010 [40]
	* Retrospective, various treatments, 4(4)	Steroids, IVIG, PP	55.2/N/A	A quarter had graft failure approximately 2 years after treatment.	Parajuli 2019 [43]
	* Retrospective, various treatments, 8(N/A)	Steroid pulse, IVIG, PP(five sessions), Belatacept	N/A/N/A	Beyond the scope of this study to discuss the optimal treatment.	Uva 2020 [46]

* Not focused on treatments; ABMR: antibody-mediated rejection; dnDSA: de novo donor specific anti-HLA antibodies; DSA: donor specific anti-HLA antibodies; IVIG: intravenous immunoglobulin therapy; PP: plasmapheresis; RCT: randomized controlled trial.

**Table 2 jcm-10-05417-t002:** Overall recurrence rate, other characteristics, available prophylaxis, and treatments in glomerulonephritis after kidney transplantation.

	IgA Nephropathy	Focal Segmental Glomerulosclerosis	Membranoproliferative Glomerulonephritis	Membranous Nephropathy
Recurrence Rate	About 30%/50–120 months (28.6%/121 ± 69 months [52], 34.9%/median 49 (range 4–213) months [53])	Differ widely between reports (10.4%/median 6.1 years (follow-up) [66], 46.7%/2.2 ± 1.8 years [92], 57.6%/median 1.25 (1 day to 30 months) months [67])	Differ widely between reports (11.8%, 15.6%, and 18.9% at 5, 10, and 15 years [70], 84.2%/76 months follow-up (C3 glomerulopathy) [82])	Differ widely between reports (10%, 16%, and 18% at 5, 10, and 15 years [70], 11.4%/median 3.6 (1.0–4.7) years [93], 44%/13.6 months [88])
Graft loss due to clinical recurrence (%)	10.8% at 10 years [53], 21.4%/130.8 ± 10.6 months follow up periods [94], 58% at 5 years [70]	Differ widely between reports (43% at 5 years [70], 39%/median 5 years [69], 9%/median 29.5 months [67])	About 50–70%/~5 years (56.3%/median 42 months (C3 glomerulopathy) [82], 70% at 5 years [70])	About 50–60%/~5 years (47.4% allograft loss/median 3.6 (1.0–4.7) years [93], 59% at 5 years [70])
Pathogenesis	Galactose-deficient IgA1, anti- galactose-deficient IgA1 IgG, immunocomplex	Circulating permeability factors, such as suPAR	Alternative complement pathway activation or immunoglobulin deposition	Anti-phospholipase A2 receptor, or thrombospondin type-1 domain-containing 7A antibodies, etc.
Risk factors of recurrence based on donor type/factors	HLA match, related donor	Related donor	Related donor (controversial)	Related donor (controversial)
Prophylaxis (Induction Therapy)	Tonsilectomy	Plasma exchange, apheresis, rituximab	Plasma exchange, rituximab	N/A
Treatments	Steroid pulse, rituximab, tonsilectomy	Plasma exchange, apheresis, rituximab	plasma exchange, rituximab, steroid pulse, or eculizumab	Steroid pulse, Rituximab

**Table 3 jcm-10-05417-t003:** Summary of recurrence rate and graft survival in patients with recurrence in lupus/ANCA/anti-GBM nephritis following kidney transplantation.

	Lupus Nephritis	ANCA Related Nephritis	Anti-GBM Abs Related Nephritis
Recurrence Rate	Vary between reports due to the frequency of biopsies, 30%/6.8 ± 4.9 (range, 3 months-20 years) [101], 4.3%/74.2 ± 72.2 months [100]	2.8% per patient year, 10%/the first 5 years post-op [105], 4.7%/median 5.5 years [106]	3.9%/median 6.4 years [109], 2.7% [110]
Graft loss due to clinical recurrence (%)	2%/6.8–4.9 (range, 3 months-20 years) [101], 1/31(3%) graft losses among 80 lupus transplant was caused by recurrence [112]	Four out of 11 recurrent cases lost theirgrafts within 5 years of transplantation [105], 2.8%/median 5.5 years [106]	3.9%/median 6.4 years [109], 0.9% [110]
Pathogenesis	Type III allergy	Neutrophil activation due to proteinase 3/myeloperoxidase-ANCA etc. [113]	Anti-GBM Abs(type II allergy)
Recommendation	Better to perform kidney transplant after introduction of dialysis therapy for several months, and being able to reduce prednisolone < 10 mg/day	Wait for renal transplant at least 12 months after the disease activity becomes under control.	Confirm a decrease in anti-GBM Abs for at least consecutive 12 months

Abs: antibodies; ANCA: anti-neutrophil cytoplasmic autoantibodies; GBM: glomerular basement membrane.

**Table 4 jcm-10-05417-t004:** Overall graft and patient survival and other key information in autoimmune liver diseases after liver transplantation.

	Primary Biliary Cholangitis	Primary Sclerosing Cholangitis	Autoimmune Hepatitis
Patient Survival after liver transplantation	About 80–90%, and 70–80% at 5 and 10 years (86% and 76% at 5 and 10 years [200], 90% and 79% at 5 and 10 years [169], 80% and 71% at 5 and 10 years [163], 84.4% and 79% at 5 and 10 years [164])	About 80–90%, and 70–80% at 5 and 10 years (78% and 70% at 5 and 10 years [163], 87.4% and 83.2% at 5 and 10 years [164], 89% and 79% at 5 and 10 years [201])	About 75% at 5 years (76–78% at 5 years [197])
Patient Survival in the recurrent group	About 95% and 80–90% at 5 and 10 years (96% and 83% at 5 and 10 years [200], 88.5%/10.1 ± 4.3 years (follow-up period) [169])	About 80%, and 50% at 5 and 10 years (84% and 56% at 5 and 10 years [201])	About 75% at 5 years (76% at 5 years [197])
Recurrence Rate	About 10% and 20–30% at 5 and 10 years(9.6% and 20.6% at 5 and 10 years [171], 13% and 29% at 5 and 10 years [200])	About 10–20% and 10–30% at 5 and 10 years (13% at 5 years [202], 14.3% at 9 years [201], 18.1% and 36% at 5 and 10 years [178], 23%/median 4.6 years [179])	About 10–20%, and 30% at 5 and 10 years (18%, and 32% at 5 and 10 years [197], 25%/15 ± 2 months (follow-up period) [203])
Pathological vs. clinical recurrence	Pathological recurrence predominant	High clinical recurrence rates (30–70%)	Pathological recurrence predominant
Prophylaxis	Ursodeoxycholic acid	N/A	N/A
Treatments	Ursodeoxycholic acid	N/A	Steroid, Azathioprine, mycophenolate mofetil
Risk factors based on donor type/factors	Gender mismatch	A first degree relative donor	HLA-DR locus mismatching, recipient DR3+/donor DR3-

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
