# Peer review of "Antibody-Mediated Rejection and Recurrent Primary Disease: Two Main Obstacles in Abdominal Kidney, Liver, and Pancreas Transplants"

_jcm, 2021, doi:10.3390/jcm10225417_

Round 1

Reviewer 1 Report

I would commend the authors on a nice overview of the field. I have some comments about issues I feel need to be addressed.

  1. I would prefer some reworking of the language and grammar as the syntax is often tortured and it affects the overall readability of the manuscript
  2. In the last para Section 2.1, ref 19 is cited correctly, then the Choi paper is not listed in the citations, and ref 20 is the Kulkarny paper on eculizumab which should have been cited prior to the Choi paper.  Ref 21 is cited correctly
  3. I was disappointed to not see reference to belatacept in DSA prevention and management. Some references noted below
    https://pubmed.ncbi.nlm.nih.gov/29509295/
    https://pubmed.ncbi.nlm.nih.gov/30582270/
  4. I would like to have seen reference to the increased risk of IgAN recurrence in steroid avoidant patients
    https://pubmed.ncbi.nlm.nih.gov/21797974/
  5. I did not think that section 3.3 was needed or at par with the rest of the manuscript. I do not consider DM as a recurrent disease in Panc Txp. I would just recommend deleting

Mainly, I think the language needs copy editing, else I liked the paper and with rework I think it merits publication.

Author Response

Response to Reviewer 1 Comments

Dear Reviewer 1.

Main Comments:
1. I would prefer some reworking of the language and grammar as the syntax is often tortured and it affects the overall readability of the manuscript. Mainly, I think the language needs copy editing, else I liked the paper and with rework I think it merits publication.

Response 1:

We greatly appreciate that the reviewer had an interest in this manuscript and added valuable suggestions as well.

Thank you for your recommendation that this manuscript be reworked in terms of language and grammar. We agree with your valuable suggestion. We amended several unnatural expressions after consultation with an English native writer.

Because we added “2. Methods” section as follows, following section numbers were passed around. e.g. former 2.1. changed 3.1.. Sorry for your inconvenience.

“We have written this review focusing on two major issues: antibody-mediated rejection (ABMR) due to de novo DSA (dnDSA) and recurrence of primary disease. In preparing this review, English-language abstracts cited in PubMed were selected. Citations were chosen based on their relevance to each section. In “3. An Overview of Antibody-mediated rejection” section, articles related with dnDSA, not preformed DSA, were selected. As recent dnDSA studies in kidney transplantation, summarized in Table 1, the most recent randomized trials or prospec-tive cohort studies using representative drugs were selected as much as possible. Since the number of related studies is scarce in liver and pancreas transplants, we selected studies which included therapeutic approaches, regardless of study design. In “4. An Overview of Recurrent Primary Disease”, we sought both studies of a relatively large scale to show reliable recurrence data and also basic mechanisms as to why the primary disease recurs, if available.”

Other Comments:
2. In the last para Section 2.1, ref 19 is cited correctly, then the Choi paper is not listed in the citations, and ref 20 is the Kulkarny paper on eculizumab which should have been cited prior to the Choi paper.  Ref 21 is cited correctly

Response 2:

Thank you for indicating our flaws in references. We corrected these issues. Now, Ref No. 20 is Kulkarni, S. et al., and No.21 is Choi, J. et al.

3.I was disappointed to not see reference to belatacept in DSA prevention and management. Some references noted below

https://pubmed.ncbi.nlm.nih.gov/29509295/

https://pubmed.ncbi.nlm.nih.gov/30582270/

Response 3:

Thank you for your insightful comments. We also believe these references are valuable in the field of ABMR, as you indicated. Considering the target patients (preformed DSA) in the latter reference, we added the former reference, which targeted at de novo DSA, in the section 3.1 (former 2.1). and Table 1.

4. I would like to have seen reference to the increased risk of IgAN recurrence in steroid avoidant patients

https://pubmed.ncbi.nlm.nih.gov/21797974/

Response 4:

Sorry for our obscure way to reference this article. To make the reference easy to find, we amended the section 4.1.1. (former 3.1.1.) as follows. Several factors may affect the pathogenesis of IgAN recurrence, such as low levels of immunosuppression, especially steroid avoidance [56] or HLA mismatch. Now, No.56 reference is this research.

5. I did not think that section 3.3 was needed or at par with the rest of the manuscript. I do not consider DM as a recurrent disease in Panc Txp. I would just recommend deleting

Response 5:

We appreciate your helpful comment. We also agree that type 2 DM is not a recurrent disease in Panc Txp. However, we also think that type 1 DM may recur after Panc Txp”. Thus, we would like to amend the 4.3. (former 3.3.) Pancreas Transplantation, with adding the following tentative explanation. “In this section, we would like to describe type 1 DM which may recur after transplant.” Then, we also would like to delete the section “4.3.2. (former 3.3.2.) Type 2 diabetes mellitus”, and keep “4.3.1. (former 3.3.1.) Type 1 diabetes mellitus”.

We wish to thank the reviewer again for your valuable comments.

Reviewer 2 Report

The authors have made impressive work reviewing the literature on ABMR and disease recurrence in the field of kidney, liver, and pancreas transplantation. This multi-organ approach is interesting but maybe a bit dispersive. Nevertheless, the manuscript is well written and intriguing.
Even if it is not a systematic review, I suggest adding a brief introductive paragraph describing the method followed for article selection.

Author Response

Response to Reviewer 2 Comments

The authors have made impressive work reviewing the literature on ABMR and disease recurrence in the field of kidney, liver, and pancreas transplantation. This multi-organ approach is interesting but maybe a bit dispersive. Nevertheless, the manuscript is well written and intriguing.

Even if it is not a systematic review, I suggest adding a brief introductive paragraph describing the method followed for article selection.

Response:

We greatly appreciate that the reviewer had an interest in this manuscript and added valuable suggestions as well. We would like to add a brief introductive paragraph the method how to select articles as follows.

“We have written this review focusing on two major issues: antibody-mediated rejection (ABMR) due to de novo DSA (dnDSA) and recurrence of primary disease. In preparing this review, English-language abstracts cited in PubMed were selected. Citations were chosen based on their relevance to each section. In “3. An Overview of Antibody-mediated rejection” section, articles related with dnDSA, not preformed DSA, were selected. As recent dnDSA studies in kidney transplantation, summarized in Table 1, the most recent randomized trials or prospec-tive cohort studies using representative drugs were selected as much as possible. Since the number of related studies is scarce in liver and pancreas transplants, we selected studies which included therapeutic approaches, regardless of study design. In “4. An Overview of Recurrent Primary Disease”, we sought both studies of a relatively large scale to show reliable recurrence data and also basic mechanisms as to why the primary disease recurs, if available.”

We wish to thank the reviewer again for your valuable comments.

Reviewer 3 Report

Major remarks:

1. The titel "abdominal transplatation" might be somehow misleading?

2. Is plasmapheresis really a B-cell targeting therapy?

3. Uva et al. added Belatacept to the maintenance immunosuppressants for selected patients, although they denied universal preemptive interventions. Can the authors also report in the results of this investigation?

4. I would ask the authors to have their manuscript checked by some professional service to increase clarity

Check spelling, grammar and/or clarity:

- the primary purpose of this review is to deepen the understanding these two issues 35 and flatten the graft survival curve after the acute phase of transplantation.

- for transplanted renal pathology

- as a major factor 80 of leading interstitial fibrosis and tubular atrophy

- Moreover, taking into consideration inflammation in IFTA area 82 may supply better prognostic views

- it is true that t

- Furthermore, worse tubulointerstitial inflammation in patients with dnDSA was more prone to recurrent 92 TCMR.[7]

- other several antibody medications

- field, although proteasome inhibitor (bortezomib) failed to show therapeutic effects

- who failed IVIG and 130 rituximab with or without plasma exchange

- does not show efficacy of these treatments.

- Many studies exist demonstrating serum dnDSA and the graft outcomes.

- ability refrain from harming liver allografts

- among total 749 liver transplant

- higher MELD score

- deteriorated or elderly patients may exhibit milder immunological reactions (deteriorated?)

- by compliment activation

- Due to the existence of hesitancy regarding pancreas graft biopsies, (e.g. Many transplant centers are hesitant to biopsy pancreas transplants etc. … the whole sentence is too long etc..

- Thus, in this review, to make it clear, recurrences (too narrative, be more precise with fewer words, don’t use wording expressing only your personal opinion too often

- Although diabetes mellitus (DM) makes up a larger percentage as a primary disease in renal transplant cases, both definitive and uncertain diagnosis with glomerular nephritis remain a significant percentage. Therefore, nephritis after transplantation cannot be recognized as recurrent or de novo nephritis precisely in these unknown cases. Do you mean type I DM? I don’t understand the meaning. Please clarify

- As a progression of αIgA research, Julian and his colleagues [59] revealed glomerular deposition of αIgA in native IgAN by means of extraction of biopsy specimens, albeit a non-individual analysis. Please clarify meaning.

Author Response

Reviewer 3.

We greatly appreciate that the reviewer had an interest in this manuscript and added valuable suggestions as well.

Major remarks:

  1. The title "abdominal transplantation" might be somehow misleading?

Response 1:

Thank you for your suggestion. To avoid misleading, we would like to change the title from "abdominal transplantation" to “kidney, liver and pancreas transplants

i.e.

“Antibody-mediated rejection and recurrent primary disease: two main obstacles in kidney, liver and pancreas transplants

  1. Is plasmapheresis really a B-cell targeting therapy?

Response 2:

We appreciate your helpful comment.  We amend this sentence using “therapies for ABMR” as follows.

In addition, several institutions seem to apply other therapies for ABMR just as in renal transplant: anti-CD 20 Abs, IVIG, and plasmapheresis

  1. Uva et al. added Belatacept to the maintenance immunosuppressants for selected patients, although they denied universal preemptive interventions. Can the authors also report in the results of this investigation?

Response 3:

Thank you for your insightful comments. Unfortunately, within our knowledge, Uva et al. briefly reported this in the discussion part without detailed information.

  1. I would ask the authors to have their manuscript checked by some professional service to increase clarity

Response 4:

Thank you for your recommendation that this manuscript be reworked in terms of language and grammar. We have amended several unnatural expressions after consultation with an English native speaker as follows (after ->).

Check spelling, grammar and/or clarity:

- the primary purpose of this review is to deepen the understanding these two issues 35 and flatten the graft survival curve after the acute phase of transplantation.

-> the primary purpose of this review is to deepen the understanding of these two issues and improve the graft survival and patient survival after the acute phase of transplants.

- for transplanted renal pathology

-> for kidney transplant pathology

- as a major factor 80 of leading interstitial fibrosis and tubular atrophy

-> as a major factor leading to interstitial fibrosis and tubular atrophy (IFTA).

- Moreover, taking into consideration inflammation in IFTA area 82 may supply better prognostic views

-> considering inflammation in IFTA area may supply better outcomes.

- Furthermore, worse tubulointerstitial inflammation in patients with dnDSA was more prone to recurrent 92 TCMR.[7]

-> Furthermore, patients with dnDSA, accompanied by more severe tubulointerstitial inflammation, were more prone to recurrent TCMR.

- it is true that t

-> We used several other expressions, such as “it must be admitted” and “it can be admitted”

- other several antibody medications

-> several other monoclonal and polyclonal antibodies

- field, although proteasome inhibitor (bortezomib) failed to show therapeutic effects

-> field, although a proteasome inhibitor (bortezomib) failed to show therapeutic effects

- who failed IVIG and rituximab with or without plasma exchange

-> for whom IVIG and rituximab, with or without plasma exchange, did not succeed.

- does not show efficacy of these treatments.

-> does not show the efficacy of these treatments.

- Many studies exist demonstrating serum dnDSA and the graft outcomes.

-> There are many studies which demonstrate serum dnDSA and the graft outcomes.

- ability refrain from harming liver allografts

-> On the other hand, other research suggests that DSA clearance by Kupffer cells, release of HLA from liver in soluble form, allografts size, and liver regenerative abilities, prevent from harming liver allografts.

- among total 749 liver transplant

-> among total 749 liver transplants

- higher MELD score

->high MELD scores

- deteriorated or elderly patients may exhibit milder immunological reactions (deteriorated?)

->deteriorating or elderly patients may exhibit milder immunological reactions

- by compliment activation

->by complement activation

- Due to the existence of hesitancy regarding pancreas graft biopsies, (e.g. Many transplant centers are hesitant to biopsy pancreas transplants etc. … the whole sentence is too long etc..

-> We separated this long sentence into two sentences as follows.

Indeed, there are criteria for ABMR of pancreas graft. Due to the hesitancy regarding pancreas graft biopsies, it is true that an accurate assessment of rejection in pancreas graft is limited on several occasions

- Thus, in this review, to make it clear, recurrences (too narrative, be more precise with fewer words, don’t use wording expressing only your personal opinion too often

“Thus, in this review, to make it clear, recurrences of the primary diseases discussed will use the following terms where possible for the sake of clarity: pathological recurrence (PR), mild-to-moderate clinical recurrence (mCR), and severe clinical recurrence (sCR) that results in end-stage organ failure.”

->

“Thus, in this review, forms of recurrences are determined as follows for the sake of clarity: pathological recurrence (PR), mild-to-moderate clinical recurrence (mCR), and severe clinical recurrence (sCR) that results in end-stage organ failure.”

- Although diabetes mellitus (DM) makes up a larger percentage as a primary disease in renal transplant cases, both definitive and uncertain diagnosis with glomerular nephritis remain a significant percentage. Therefore, nephritis after transplantation cannot be recognized as recurrent or de novo nephritis precisely in these unknown cases.

Do you mean type I DM? I don’t understand the meaning. Please clarify

Response:

Sorry for this confusing expression. We tried to explain that type 2 DM is a major reason for renal failure and accounts for a larger percentage. However, we suppose that the part form “Although…. Transplant cases” is not necessary. Thus, we would like to change these sentences as follows.

“There are many renal transplant recipients whose primary diseases are uncertain. In these cases, nephritis after transplantation cannot be recognized as recurrent or de novo nephritis precisely.

- As a progression of αIgA research, Julian and his colleagues [59] revealed glomerular deposition of αIgA in native IgAN by means of extraction of biopsy specimens, albeit a non-individual analysis. Please clarify meaning.

Response:

Thank you for your insightful comments. This research used pooled samples, collected from 4 or 5 patients, simply because the physical size of specimens was not enough. To avoid confusion, we would like to change this sentence as follows.

“As a progression of αIgA research, Julian and his colleagues [59] revealed glomerular deposition of αIgA in native IgAN by means of extraction of biopsy specimens, although not having enough tissue prevented an individual analysis.

We wish to thank the reviewer again for your valuable comments.